# Research on Environmental Risk Monitoring and Advance Warning Technologies of Power Transmission and Distribution Projects Construction Phase

**DOI:** 10.3390/s24237695

**Published:** 2024-12-01

**Authors:** Xiaohu Sun, Fei Liu, Yu Zhao, Fang Liu, Jian Wang, Shu Zhu, Qiang He, Yu Bai, Jiyong Zhang

**Affiliations:** 1State Grid Economic and Technological Research Institute Ltd., Beijing 102200, China; 13810318668@163.com (X.S.); zhushu9293@163.com (S.Z.); myinternetcn@163.com (J.Z.); 2School of Geomatics and Urban Spatial Information, Beijing University of Civil Engineering and Architecture, Beijing 102616, China; liufei1@bucea.edu.cn (F.L.); wangjian@bucea.edu.cn (J.W.); 3School of Science, Beijing University of Civil Engineering and Architecture, Beijing 102616, China; heqiang@bucea.edu.cn (Q.H.); baiyu@bucea.edu.cn (Y.B.)

**Keywords:** power transmission and distribution project, environmental monitoring and supervision technologies, unmanned aerial vehicle (UAV), multi-sensor fusion, risk atlas, advance warning technology

## Abstract

The threat power transmission and distribution projects pose to the ecological environment has been widely discussed by researchers. The scarcity of early environmental monitoring and supervision technologies, particularly the lack of effective real-time monitoring mechanisms and feedback systems, has hindered the timely quantitative identification of potential early-stage environmental risks. This study aims to comprehensively review the literature and analyze the research context and shortcomings of the advance warning technologies of power transmission and distribution projects construction period using the integrated space–sky–ground system approach. The key contributions of this research include (1) listing ten environmental risks and categorizing the environmental risks associated with the construction cycle of power transmission and distribution projects; (2) categorizing the monitoring data into one-dimensional, two-dimensional, and three-dimensional frameworks; and (3) constructing the potential environmental risk knowledge system by employing the knowledge graph technology and visualizing it. This review study provides a panoramic view of knowledge in a certain field and reveals the issues that have not been fully explored in the research field of monitoring technologies for potential environmental damage caused by power transmission and transformation projects.

## 1. Introduction

Electricity serves as a crucial lifeline for human energy security and the global economy. The construction of power systems plays a pivotal role in alleviating regional energy development disparities, promoting essential living infrastructure across regions, and providing vital guarantees for regional energy security, economic growth, and public well-being.

This study focuses on the technological progress of the potential risks that engineering construction poses to the environment throughout the entire lifecycle of transmission and distribution (T&D) project construction. Generally, power systems comprise power generation, transmission and distribution, and distribution projects. To ensure the minimization of environmental impacts during the construction and operation of transmission and distribution projects globally, Table 1 presents the regulatory bodies and industry-standard advancements in this domain at the international level in the United States, the European Union, and China. Internationally, the International Energy Agency (IEA) and the International Electrotechnical Commission (IEC) have released numerous reports on the environmental impact assessment of global power transmission and distribution networks. In the United States, the technology reports, regulations, and measures for environmental impact assessment of transmission and distribution projects are primarily formulated and managed by organizations such as the U.S. Environmental Protection Agency (EPA), American Electric Power (AEP), and the Electric Power Research Institute (EPRI). The EU’s specifications for transmission and distribution projects encompass multiple organizations and regulatory frameworks, including the European Environment Agency (EEA) and the European Commission (EC), providing assessment techniques and measures to prevent and restore environmental damages. In China, the formulation of environmental impact assessments (EIA) and related standards for transmission and distribution projects is collaboratively undertaken by the National Energy Administration (NEA), the Ministry of Ecology and Environment (MEE), and the China Electricity Council (CEC). This entails establishing an internal review mechanism for EIA reports during the feasibility study phase, advocating the concept of “integrated ecological environmental protection and design” during the design phase, actively applying new technologies, materials, and processes during construction, strengthening environmental and water conservation supervision on-site, and clarifying acceptance criteria and standardizing work procedures during the completion stage, thereby minimizing the adverse effects of the entire grid construction process on the surrounding environment. However, any construction project, including transmission and distribution projects, inevitably involves excavation and consequently impacts the ecological environment. Table 2 summarizes the construction stages of transmission and transformation projects and their impacts on the environment. For instance, during the construction preparation stage, clearing the construction site may involve cutting down trees and damaging vegetation [1,2]. Earthworks at this stage can lead to water and soil erosion, the accumulation and pollution of solid waste materials, wastewater discharge, and wind erosion of the soil [3,4,5,6]. Furthermore, during the tower foundation excavation phase, the destruction of vegetation cover may potentially cause landslides and debris flows. In the tower erection and wire stringing stages, noise and dust generated by the operation of construction machinery and equipment may adversely affect nearby residents and wildlife [7,8]. Additionally, during the substation installation phase, large amounts of mud and construction wastewater may be discharged into local water bodies, causing water pollution. Once operational, high-voltage transmission lines and substations can generate a certain level of electromagnetic radiation, potentially affecting the surrounding environment and the health of local residents. Moreover, the daily operation of substations may produce continuous low-frequency noise, which could also affect the environment and public health. Similarly, the transmission and distribution system may contribute to the fragmentation of ecological green spaces.

The potential risks that engineering construction poses to the environment throughout the entire lifecycle of transmission and distribution project construction primarily refer to transmission and distribution substations, towers, and lines. These risks may include but are not limited to tower foundation disturbance and restoration, changes in vegetation cover, monitoring of water and soil erosion, impacts of engineering measures, construction and management of temporary facilities, relocation and resettlement of residents, control of construction noise, management of dust, waste disposal and resource utilization, as well as ecological restoration. Traditional methods of environmental risk monitoring primarily rely on manual labor, which is characterized by low accuracy, high workload, and time-consuming nature, being unable to monitor large-scale environmental changes during the construction period of power transmission and distribution projects [49,50]. Currently, technologies such as 5G, unmanned aerial vehicles (UAVs), BeiDou Navigation Satellite System, artificial intelligence, and digital twins are accelerating their in-depth integration with industrial sectors like power grids. The utilization of remote monitoring and UAV-based monitoring as new technologies is gradually becoming prevalent [51,52,53]. Remarkable advancements have been made in risk monitoring and advance warning technologies. Through the integration of various high-precision sensor devices mounted on ground equipment, UAVs, and satellites, comprehensive one-dimensional, two-dimensional, and three-dimensional data collection and analysis are achieved. The integration and application of these technologies have significantly enhanced the precision and efficiency of environmental monitoring during the construction of power transmission and distribution projects. This enables the timely detection and diagnosis of environmental impacts and damage during the construction period, informing construction parties to adopt comprehensive measures to mitigate the impacts on the surrounding environment, safeguarding public health, and protecting the ecological environment.

Based on the aforementioned advancements in risk monitoring and advance warning technologies, the novelty and contributions of this review are manifested in the following aspects:(1)We listed ten environmental risks and categorized them according to the entire lifecycle of transmission and distribution engineering construction, providing a basis for subsequent monitoring and advance warning efforts.(2)The monitoring data of risk were categorized into one-dimensional, two-dimensional, and three-dimensional frameworks in order to make data management and processing clearer and easier.(3)We constructed an environmental risk knowledge system by employing the knowledge graph technology. Moreover, we proposed the environmental risk advance warning system.

This paper delves into the research on environmental risk monitoring and advance warning technologies during the construction period of power transmission and distribution projects. Section 1 serves as an introduction, outlining the background of environmental risk monitoring and advance warning technologies for such projects. Following this, Section 2 systematically collects and analyzes the most relevant research findings on these technologies, laying a solid foundation for subsequent investigations. Section 3 comprehensively lists the primary environmental risk factors encountered during the construction period, organized by the construction preparation stage, construction stage, and ecological restoration and periodic inspection stages, thereby clarifying the targets for subsequent monitoring and advance warning efforts. Section 4 elaborates on the current state of data acquisition and collection technologies for risk factors, exploring the opportunities and challenges of each technology in practical applications. Section 5 delves into the data governance processes for environmental risk factors. Section 6 innovatively proposes a method for constructing an environmental risk factor map and visually represents the distribution of environmental risks during the construction period through visualization. Section 7 focuses on risk advance warning technologies to explore the utilization of advanced techniques for real-time monitoring and advance warning of environmental risks, ensuring the safety and environmental protection of power transmission and distribution projects. Section 8 concludes the review with a summary and outlook, we summarize the key research findings of this paper and envision the future directions and application prospects of environmental risk monitoring and advance warning technologies during the construction period of power transmission and distribution projects. The structure of the article is illustrated in Figure 1.

## 2. Retrieval Strategy

Comprehensive retrieval was conducted using the databases of Web of Science, IEEE Xplore, ScienceDirect, Google Scholar, and Ei Compendex. As shown in Table 3.

**A** **Inclusion and exclusion criteria**

Articles will be included in this study if they meet the following criteria:(1)Those that are based on the risk types associated with the construction period of power transmission and distribution projects.(2)Those that introduce various environmental risk monitoring methods.(3)Those that describe multi-sensor fusion technologies.(4)Those that focus on the construction and advance warning systems based on environmental risk atlas.(5)Those for which the full text is accessible and readable.

Articles will be excluded from this study if they meet any of the following criteria:(1)Those published in languages other than English.(2)Those appearing in low-quality publications or venues, i.e., publications or venues outside the databases in Table 3.(3)Those for which the full text is inaccessible.

**B** **Data extraction and analysis**

The first step in the selection process involved screening through titles, abstracts, and introductions to determine the level of relevance of each paper. Subsequently, the exclusion criteria outlined in Section A were applied to eliminate papers that did not meet the inclusion criteria. From each of the included articles, relevant information was extracted, encompassing authors, publication source, year of publication, methodological approach, datasets utilized, evaluation metrics, and conclusions. As shown in Table 4.

## 3. Types of Environmental Risks

The environmental risk types intimately associated with the full lifecycle of power transmission and distribution projects construction can be categorized into three stages: construction preparation stage, construction phase, and regular inspection stage. The primary ecological and environmental risk types encountered during these three stages of power transmission and distribution project construction include tower foundation disturbance, vegetation cover changes, water and soil erosion, residential house demolition, construction noise and dust, waste disposal and resource utilization, impacts of engineering measures, temporary facility construction and management, as well as ecological restoration, among others. These are the primary targets of our monitoring efforts, as illustrated in Figure 2. For instance, during the construction preparation stage, environmental issues may arise from route selection and planning, land acquisition, residential relocation, and the creation of new access roads [18,38,54,55,56]. During the construction phase, the construction of transmission substation towers, substations, and transmission lines inevitably involves the use of large-scale machinery and equipment, which may further contaminate and damage the land, vegetation, water sources, and air at the construction site [57,58].

### 3.1. Construction Preparation Stage

Throughout the entire lifecycle of power transmission and distribution project construction, the construction preparation phase is an indispensable component. Activities such as road clearance and development, residential house demolition, material transportation and stacking, as well as the mobilization of machinery and equipment, primarily pose ecological and environmental risks centered on changes in vegetation cover, water and soil erosion, and potential environmental pollution arising from residential demolition (Table 2).

The construction preparation phase of power transmission and distribution projects can lead to multifaceted environmental issues [59,60]. During site clearance and leveling, the creation of new roads and the clearing of construction sites require the removal of extensive vegetation, potentially leading to ecosystem disruption, increased risks of soil erosion, and the loss of local biodiversity [1,2,61,62]. Furthermore, the reduction in vegetation cover accelerates the risk of soil erosion. The soil surface, originally protected by vegetation, becomes vulnerable to wind and water erosion after vegetation destruction, resulting in soil loss and degradation. Additionally, earthworks conducted during the preparation phase (such as excavation, filling, and grading) often expose large areas of soil, making them susceptible to rainfall washout and wind erosion [3,63,64,65]. Especially around construction sites, the soil structure and stability are prone to damage due to extensive machinery operations and human activities, thereby intensifying the risk of soil loss [4,66,67]. Moreover, due to soil erosion, significant amounts of sediment and water containing pollutants may enter nearby water bodies, leading to water quality degradation and sedimentation issues [68,69,70,71,72,73]. Furthermore, during the preparation phase of power transmission and distribution projects construction, the demolition of residential houses generates vast quantities of construction waste. If the waste storage sites are not properly planned and managed, they may contaminate surrounding soil and water bodies. During the demolition process, the destruction and clearance of buildings can generate considerable dust emissions, which contain harmful substances such as heavy metals and volatile organic compounds, posing hazards to the surrounding environment and human health. Additionally, the use of machinery and the sounds of collapsing buildings create noise disturbances for nearby residents, affecting their quality of life and health [7,12,74].

### 3.2. Construction Stage

During the construction phase of power transmission and distribution projects, ecological and environmental risks primarily center around tower foundation disturbance, construction noise and dust, water and soil erosion, waste disposal and resource utilization, damage to soil and vegetation, as well as the environmental impacts stemming from engineering and temporary measures (Table 2).

In the construction of transmission and distribution towers’ foundations, a series of engineering measures such as earthwork excavation and filling inevitably lead to environmental issues like construction noise and dust. The excavation process for tower foundation pits disrupts vegetation, potentially causing severe water and soil erosion during rainy or windy weather [3,19,20,75]. The utilization of various mechanical equipment (such as excavators, bulldozers, and drilling machines) and transport vehicles during construction generates high-intensity noise. This noise not only affects the health of construction personnel but also disrupts the lives of nearby residents. Additionally, it disturbs the lives and breeding of local wildlife [76,77,78,79,80,81]. Extensive earthwork excavation, transportation, loading and unloading, along with the utilization of construction materials (such as cement and gravel), generates substantial dust. This dust not only degrades the air quality at construction sites and in surrounding areas but also poses hazards to the respiratory systems of construction workers and nearby residents. High concentrations of dust can also reduce visibility, increasing traffic risks [8,82,83,84,85]. Large-scale land leveling, excavation, and the construction of access roads disrupt the original vegetation cover, resulting in a decrease in green space [4,86]. Furthermore, the disturbance of the earth’s surface during construction activities (such as excavation, grading, and mechanical compaction) disrupts soil structure, enhancing the risk of water and soil erosion. The exposed earth’s surface is susceptible to erosion by wind and rain, leading to decreased soil fertility and sediment accumulation in rivers [87,88,89,90]. Moreover, substantial amounts of construction waste, used packaging materials, discarded equipment parts, and other solid waste are generated during the construction of transmission towers and substations. If these wastes are not promptly and effectively disposed of, they may accumulate at the construction site or in surrounding areas, causing environmental hygiene issues. Construction waste may contain hazardous substances that, if not properly managed, can contaminate soil and water bodies [28,33,34,35,36]. Additionally, oil, chemicals, and other pollutants that may leak during construction can also contribute to soil contamination [91,92]. Furthermore, the temporary stockpiling of large quantities of construction materials, earthwork, and waste during construction necessitates careful management. Poorly managed stockpiling sites may lead to environmental issues such as dust emissions, sewage seepage, and contamination of soil and water bodies. During rainfall, stockpiled earthwork is particularly prone to erosion, exacerbating the burden on the surrounding environment [28,39,93]. The erection of temporary construction facilities (such as sheds and warehouses) may consume land resources, alter local landforms and water flow patterns, and increase the risk of water and soil erosion.

### 3.3. Regular Inspection Stage

Upon the completion of power transmission and distribution projects, it is essential to conduct an inspection and evaluation of the relevant ecological restoration measures adopted during the construction process, primarily focusing on the restoration of ecological footprints, such as vegetation rehabilitation and soil erosion control (Table 2). A comprehensive ecological restoration inspection and evaluation not only contributes to environmental protection but also provides valuable experience and improvement suggestions for future construction projects, thereby enhancing the overall sustainability of the project.

During the entire lifecycle of power transmission and distribution projects, it is crucial to assess whether the temporary construction roads have been demolished according to plan and whether the surface has undergone restoration, including land leveling and removal of residual construction materials. Additionally, it is necessary to verify if the originally damaged roads have been repaired and restored to their pre-construction condition, encompassing aspects such as pavement evenness and the integrity of drainage facilities. Regarding vegetation restoration in the construction area, the planting status of turf, shrubs, trees, and other vegetation should be evaluated to ascertain whether it aligns with the ecological restoration plan [43,59,94]. Furthermore, it is imperative to ascertain whether necessary soil and water conservation measures, such as vegetation-covered slope protection, retaining walls, and drainage ditches, have been implemented in the construction area [20,45]. Finally, a comprehensive assessment of the ecosystem recovery in the construction area should be conducted, encompassing soil quality, vegetation coverage, wildlife species diversity, and population size [95,96].

UAVs have become one of the main tools for ecological monitoring of power transmission and distribution projects. By carrying high-definition cameras and multispectral and hyperspectral sensors, it can quickly obtain ecological monument restoration measures such as vegetation restoration and road restoration [96,97]; and by laying sensors in ecological restoration areas, it can collect real-time environmental data such as soil humidity, temperature, water quality, and so on, which facilitates long-term monitoring and dynamic assessment [98,99]. These methods have been widely used in dynamic monitoring of vegetation restoration in transmission line construction areas, featuring real-time and high accuracy and high technological maturity, but they are still limited by weather conditions and battery life, and the coverage and maintenance costs of sensor deployment are still challenges.

## 4. Advances in Data Monitoring Technology

In recent years, there have been significant advancements in data acquisition and collection devices and technologies for monitoring environmental risks during the construction phase of transmission and distribution projects. These advancements are primarily reflected in innovations in sensor technology, such as the development of smart devices equipped with various sensors, including infrared cameras, hyperspectral cameras, video cameras, and LiDAR (light detection and ranging). This section will elaborate on the development of data acquisition devices and technologies related to environmental risk factors throughout the transmission and distribution project construction cycle. This includes one-dimensional time series and spatial data, such as noise and dust data acquisition; two-dimensional video and image data, encompassing but not limited to vegetation change, soil erosion, residential house demolition, ecological restoration, engineering measures, and temporary facilities; and three-dimensional model data, including but not limited to data acquisition for tower foundation disturbance, vegetation change, and soil erosion. Figure 3 illustrates the proposed integrated space–sky–ground system for environmental risks during the construction period of power transmission and distribution projects.

### 4.1. One-Dimensional Signal

The one-dimensional signal is a scalar function that varies with an independent variable (such as time or space), and its core property is to record the dynamics of a single variable in a certain dimension. In our study, the one-dimensional data involved are mainly time series data of noise and dust, which are mainly concerned with the intensity or value that changes over time in the time dimension, such as the decibel value of noise and the concentration of dust. However, when the time series is analyzed in conjunction with additional characteristic variables (such as spatial location, wind speed, ambient temperature, etc.), the signal can essentially be mapped to a two or even higher-dimensional characteristic space, such as when we analyze the correlation between the dust concentration with the wind speed or temperature and humidity in a time series, the data can be represented as a two or even multi-dimensional signal. Noise and dust data, characterized by continuous time series data collection through sensors and single measurement dimensions, serve as a typical application of one-dimensional signal processing. Activities such as residential demolition and new road construction during the preparatory phase, as well as the construction of transmission towers, substations, and transmission lines during the construction phase, generate noise and dust. Monitoring these environmental parameters can be effectively accomplished using mobile monitoring devices or ground-based monitoring stations equipped with noise and dust sensors. These devices offer flexible coverage of construction areas, providing efficient and accurate real-time environmental data, which aids in the timely identification and control of pollution sources [100,101,102,103]. Noise sensors determine noise intensity and measure noise levels by capturing and analyzing acoustic wave signals in the environment. Dust sensors, on the other hand, utilize laser scattering technology to detect the concentration of particulate matter (such as PM10 and PM2.5) in the air [104]. The data collected by these sensors are transmitted in real-time to a data processing system for analysis and storage. Mobile devices enable dynamic monitoring by moving through the construction area [100,102]; while ground-based fixed devices are stationed at specific locations for continuous monitoring [101,103]. The method is straightforward, easy to deploy, and effectively meets the environmental monitoring needs during the construction phase of power transmission and distribution projects.

With the innovation of sensor technology and the intelligentization of carrier devices, noise and dust monitoring technologies have continued to advance, facilitating the monitoring of environmental risk factors such as noise and dust during the construction of power transmission and distribution projects. Traditional noise monitoring primarily relied on handheld noise meters, which, while capable of acquiring noise data from localized areas, were limited in their measurement capabilities to specific time points, data points, and coverage and thus unable to achieve real-time monitoring [105]. Noise and dust data, as one-dimensional signals, are primarily analyzed through time series data, enabling real-time reflection of environmental changes during construction. Early noise monitoring devices primarily relied on simple decibel meters, involving manual recording of noise data, which suffered from issues like low data acquisition efficiency and poor accuracy. With technological advancements, the application of digital noise meters and intelligent sensors has substantially improved the automation and precision of data collection. Modern noise monitoring equipment integrates wireless transmission modules, enabling real-time data uploading for remote monitoring and analysis [106,107]. The development of dust monitoring equipment has undergone a transformation from simple optical dust meters to comprehensive monitoring systems integrating multiple sensors [108]. Initial dust monitoring devices could only roughly measure the concentration of particulate matter in the air, with relatively poor data accuracy and stability. With the introduction of light scattering technology, laser detection technology, and electrochemical sensors, modern dust monitoring equipment is now capable of detecting particles of different sizes with greater precision, providing high-resolution real-time data. These devices not only possess high accuracy and sensitivity but also enable remote data management and intelligent analysis through cloud platforms. Nevertheless, current technologies still exhibit certain limitations, such as the complexity of data processing and analysis, high equipment costs, and stability issues under extreme environmental conditions. Future development directions include higher-performance sensors, more intelligent data analysis methods, and more convenient remote monitoring and management platforms, aiming to comprehensively enhance the efficiency and reliability of environmental monitoring during engineering construction.

### 4.2. Two-Dimensional Image

Remote sensing imagery data acquired by drones or satellite devices, including visible light, near-infrared, and shortwave infrared spectral information, exhibit two-dimensional spatial distribution characteristics, encompassing both horizontal and vertical dimensions. Therefore, sensors mounted on drones or satellites (such as infrared cameras, hyperspectral cameras, and video cameras) can be employed for vegetation change and ecological restoration monitoring. These sensors efficiently and accurately cover extensive construction areas during the transmission and distribution project phases, providing real-time, comprehensive ecological data that facilitates timely assessment and management of ecological restoration progress [39,44,109]. Drones or satellites capture spectral reflectance data and multispectral remote sensing images of surface vegetation to analyze vegetation coverage and the progress of ecological restoration. The video and image data collected by the sensors are processed through remote sensing techniques to generate vegetation indices (such as the Normalized Difference Vegetation Index, NDVI) and ecological restoration evaluation reports. Infrared cameras capture vegetation indices within construction areas, quantifying the health status of vegetation [110,111]. Hyperspectral cameras, through spectral reconstruction, obtain near-continuous spectral reflectance data of surface features, enabling the identification of plant species and the calculation of the NDVI [112,113]. Video cameras provide high-resolution visible light images [114,115]. UAVs can fly at low altitudes, providing high-resolution local detail data, while satellites offer broader coverage, making them suitable for long-term monitoring sequences [116]. UAVs can fly at low altitudes, providing high-resolution local detail data, while satellites offer broader coverage, making them suitable for long-term monitoring sequences. Upon transmission to ground stations, the data undergoes further processing and analysis to evaluate ecological indicators such as vegetation recovery and soil moisture. This monitoring method is efficient and convenient, ideal for large-scale ecological restoration monitoring and assessment. Corresponding research also exists in monitoring environmental risk factors during the construction of power transmission and distribution projects, including water and soil erosion, residential house demolition, engineering measures, and temporary measures [117,118,119].

The development of data acquisition technologies in drone and satellite remote sensing has significantly enhanced the spatial resolution, temporal resolution, and data processing capabilities for monitoring construction areas. The techniques for monitoring vegetation change and ecological restoration based on image or video data have continuously advanced. Traditional monitoring methods predominantly relied on manual ground surveys and sample analysis, which, despite providing detailed vegetation information and data on ecological restoration measures, were limited in coverage, time-consuming, and costly [120]. The development of modern technologies has transformed the multispectral, hyperspectral, and infrared cameras mounted on UAVs and satellites into primary tools for change monitoring. For example, in one of China’s most forested areas, the upper reaches of the Minjiang River in Fujian Province, ref. [109] used Landsat multispectral imagery and forest inventory datasets to quantitatively analyze the impact of high-voltage transmission line construction on forest landscape patterns and vegetation growth, thereby monitoring vegetation change in the construction area. The data showed that the construction of high-voltage transmission lines occupied a significant proportion of forest land, which could potentially have adverse effects on wildlife habitats and wildfire management. These data provide real-time and accurate reflections of vegetation changes, aiding in the timely implementation of protective and restorative measures. Ref. [39] utilized five sets of satellite multispectral sub-meter resolution remote sensing data to study vegetation changes in the construction area of the Jixi Pumped Storage Power Station in Anhui. This approach allowed me to obtain vegetation coverage at different construction stages, which was then used to evaluate the coverage of post-construction artificial vegetation restoration measures.

In the context of environmental protection monitoring during power transmission and distribution project construction, technological advancements in two-dimensional image acquisition and collection devices are of paramount importance. Satellite remote sensing technology, with its advantages of wide coverage, short monitoring cycles, and high spatial resolution, enables the periodic acquisition of high-quality image data over vast areas. This capability can be harnessed to monitor changes in vegetation cover and ecological restoration during the construction phase of power transmission and distribution projects [121]. UAVs are capable of flying at low altitudes to capture high-resolution real-time imagery, making them suitable for detailed and localized environmental monitoring. Furthermore, UAVs offer the advantages of rapid deployment and reusability, rendering them ideal for emergency monitoring in response to incidents as well as for regular inspections [122,123]. However, it is noteworthy that the relevant data acquisition equipment is constrained by the limited penetration power of optical imagery, posing challenges in obtaining high-quality data under dense vegetation cover or harsh weather conditions.

### 4.3. Three-Dimensional Data

LiDAR is a sampling tool that precisely calculates the position of target objects in three-dimensional space by measuring the time delay and intensity of laser pulses reflected from the objects. The data collected by LiDAR includes not only horizontal and vertical two-dimensional information but also height (or depth) information, thereby forming a complete three-dimensional point cloud dataset. Oblique photography is a technique that uses multiple angles to capture images, which is widely used in three-dimensional modeling and geographic information acquisition and is particularly suitable for large-scale surface restoration, building monitoring, ecological restoration, and other fields [124,125]. Unlike traditional vertical aerial photography, oblique photography can provide a more comprehensive view and richer spatial information by capturing images of the same target area from multiple directions. Compared to two-dimensional image data, LiDAR can more accurately locate and measure the position and morphology of target objects. In vegetation change and ecological restoration monitoring, it captures vegetation structure, terrain features, and environmental changes in greater detail. By utilizing LiDAR scanning technology mounted on drones or satellites, vegetation change and ecological restoration in transmission and distribution construction areas can be monitored efficiently and accurately, supporting comprehensive and detailed ecological assessments, which is particularly suitable for environmental risk monitoring during the construction of transmission and distribution projects [109,126,127]. The data are processed and analyzed through remote sensing technology, generating three-dimensional images and reports that include vegetation height, density, and coverage, used to assess the impact of construction on vegetation and the effectiveness of ecological restoration. This monitoring method is efficient and precise, making it suitable for large-scale and complex terrain ecological monitoring. It has been applied in the status monitoring of transmission towers, substations, and line construction, as well as in monitoring the process of residential house demolition [128,129]. Table 5 compares the advantages and disadvantages of different environmental factor data acquisition methods

Vegetation change and ecological restoration monitoring technologies have continuously advanced. From initial ground surveys and manual records to video and image data acquired by sensors mounted on drones and satellites, and now to the adoption of LiDAR scanning technology, these monitoring methods have seen significant improvements in efficiency and accuracy. Ref. [44] used satellite-mounted LiDAR equipment to monitor vegetation in the construction area of a 22KV transmission network in rural western Norway. LiDAR data can reflect vegetation structure changes in real time and with high precision, aiding in the timely implementation of protective and restorative measures.

In the ecological and environmental monitoring during the construction phase of transmission and distribution projects, LiDAR technology is particularly advantageous for acquiring three-dimensional data on vegetation cover changes and ecological restoration [130]. Initially, three-dimensional data acquisition primarily relied on traditional ground-based survey methods, which, while accurate, were inefficient, limited in coverage, and inadequate for large-scale and complex terrain monitoring needs. Modern LiDAR systems have since been implemented across various platforms, including airborne, vehicle-mounted, and ground-based systems. Airborne LiDAR, mounted on aircraft or drones, enables rapid scanning of extensive areas, capturing detailed three-dimensional terrain and vegetation structure information. This is especially useful for monitoring changes in vegetation cover and identifying potential risk points along transmission line corridors. Vehicle-mounted LiDAR is employed for detailed ground measurements, capable of capturing the three-dimensional shapes of roads, buildings, and other ground objects, which are essential for precise measurement and management during the construction process. Ground-based LiDAR is suitable for detailed local environmental monitoring, such as the detailed analysis of tower foundations and construction sites. However, the limitations of LiDAR technology are also apparent. High-precision LiDAR equipment is expensive, and the costs are even higher for large-scale high-precision scanning tasks, requiring substantial equipment and technical support. The operation and deployment of this equipment necessitate professional technical personnel, particularly in complex terrain and variable environmental conditions.

**Table 5 sensors-24-07695-t005:** Compares the advantages and disadvantages of different environmental factor data acquisition methods during the construction phase of transmission and distribution projects.

Monitoring Methods	Disadvantages	Advantages	Related Research
Ground-based monitoring equipment	Poor real-time performance with limited coverage.	Portability and ease of operation.	For instance, noise and dust monitoring [131,132,133].
Aerial monitoring equipment	Highly influenced by weather conditions, limited endurance, and expensive.	Offers real-time monitoring, good data continuity, and high image resolution.	For instance, monitoring vegetation changes, soil erosion, and engineering measures [134,135].
Satellite monitoring equipment	Low image resolution and poor data continuity.	Offers extensive coverage and is suitable for long-term sequential monitoring.	For instance, monitoring vegetation changes and soil erosion [109,136,137].

### 4.4. Multi-Source Data Fusion

In the data collection process for risk factor monitoring during the construction phase of transmission and distribution projects, many existing studies that use remote sensing data for change detection have relied on single sensors or images [113]. This approach may fall short of meeting the complex application needs, such as detecting various types of risks. Multisource data refers to the integration of data from different sources to enhance the potential value of the source data and improve its visualization, including optical imagery, LiDAR, GIS data, and ground truth data [138]. Multitemporal data involves a time series of multiple images or at least a pair of images from the same geographic study area taken at different times [139]. For instance, in some studies on vegetation changes, multispectral images combined with LiDAR data have been employed for more accurate forest classification [140]. The integration of remote sensing data with GIS data (such as topography, land use, and census data) can enhance the accuracy of target identification, image classification, and change detection. This fusion not only improves the spatial and temporal resolution of the data but also enables a more precise assessment of environmental change trends and impacts, providing a scientific basis for environmental management and ecological restoration decisions. The continuous optimization and maturation of multisource data fusion techniques are becoming crucial tools in environmental monitoring for transmission and distribution projects, facilitating environmental protection and management during the construction phase and throughout the project lifecycle.

## 5. The Governance of Environmental Risk Data

The advancement of data analysis and processing techniques is equally crucial in environmental risk monitoring during the construction phase of transmission and transformation projects. This section focuses on the progress in the analysis and processing of risk factor data, acquired as discussed in the previous section, to elaborate on the technical advancements in environmental risk factors monitoring during the construction phase of transmission and distribution projects. Figure 4 illustrates the main technical processes of this subsection.

### 5.1. Preprocessing

In the context of environmental risk monitoring during the construction phase of transmission and distribution projects, data preprocessing is essential to ensure the accuracy and consistency of data obtained from various devices and sensors. This process involves the standardization of multi-temporal, multi-sensor, or multi-view imagery data, making it more suitable for subsequent analyses. For one-dimensional data such as noise and dust, the preprocessing steps include denoising, outlier detection and removal, time synchronization, and standardization. These steps are crucial for maintaining the accuracy and consistency of the data. For three-dimensional deformation data obtained from LiDAR, preprocessing involves several steps, such as data decompression, point cloud denoising, point cloud registration, coordinate transformation, geometric correction, meshing, and standardization. These procedures ensure the reliability, completeness, and consistency of the data, providing a solid foundation for further analysis. Given that the environmental risk factor data collected during the construction of transmission and distribution projects are primarily video and imagery data, this subsection focuses on the advancements in preprocessing techniques for such data.

The preprocessing of two-dimensional video and imagery involves steps such as decompression, atmospheric correction, geometric correction, image-to-image registration, radiometric calibration, and orthorectification to obtain preprocessed data [39]. Advancements in related research include the following: Decompression, for instance, refers to restoring compressed image data to processable raw form. Efficient lossless and lossy compression algorithms, from early JPEG and LZW formats to modern JPEG2000 and HEVC formats, have significantly improved decompression speed and data recovery quality, enabling faster and more accurate processing of large-scale remote sensing imagery data [141,142,143,144,145,146,147]. Atmospheric correction is the process of removing radiometric distortions in imagery caused by atmospheric scattering and absorption, thus restoring the true reflectance of surface features. Early methods, such as the Dark Object Subtraction (DOS) technique [148], have evolved into more sophisticated models, like the Second Simulation of a Satellite Signal in the Solar Spectrum (6S) [149,150,151]. Similarly, geometric correction is employed to eliminate geometric distortions in imagery caused by factors such as sensor perspective, terrain undulations, and motion, thereby aligning the imagery within a geographical coordinate system [152]. This field has advanced from early affine transformation methods to modern three-dimensional geometric correction techniques based on Digital Elevation Models (DEMs), which offer greater precision [153,154,155]. Lastly, image registration involves the spatial alignment of multi-temporal, multi-sensor, or multi-view imagery to remove spatial discrepancies between images [156]. Recent developments in this area include the use of convolutional neural networks (CNNs) for feature extraction and matching, combined with large-scale data training, significantly enhancing the accuracy and efficiency of image registration [157]. Ref. [158] used ArcGIS software to pre-process lidar data, orthophotos data, and satellite data to generate a subset of images of the study area of Forest Research Institute Malaysia forest. The aim was to use the geospatial data to make recommendations for optimizing the siting of transmission lines in tropical forests to avoid endangered tree species.

In the context of environmental risk factor monitoring during the construction of transmission and distribution projects, the preprocessing of multi-temporal imagery data faces challenges related to time synchronization and seasonal variations. Imagery acquired at different times may exhibit radiometric inconsistencies due to varying lighting conditions and weather changes, complicating radiometric and atmospheric correction. Furthermore, the fusion of multi-sensor data is challenging due to differences in spatial resolution, spectral resolution, and geometric properties, making geometric correction, image registration, and orthorectification more complex. Preprocessing multi-view imagery also requires addressing geometric distortions caused by varying viewing angles, necessitating more precise geometric correction and registration techniques to ensure data consistency and accuracy.

### 5.2. Extraction of Characteristics of Environmental Risk Factors

In the environmental risks monitoring during the construction period of power transmission and distribution projects, feature extraction is employed to distill crucial information from vast amounts of environmental data, encompassing vegetation cover changes, soil erosion, noise levels, dust emissions, and the like. Its merits lie in swiftly and precisely identifying and assessing environmental variations and potential risks, thereby enhancing the efficiency and reliability of environmental monitoring.

By analyzing two-dimensional data imagery, including optical images, videos, multispectral, hyperspectral, radar, and thermal infrared images, the impacts of construction activities on the environment can be effectively identified and assessed. Given the continuous advancements in network models such as machine learning and deep learning, the processing of two-dimensional videos and images has become more convenient and rapid. For the monitoring of vegetation changes in the construction area of power transmission and distribution projects, the use of aerial images or satellite images, combined with deep learning algorithms, achieves the automatic detection and classification of vegetation cover areas. The areas of vegetation reduction and change trends can be accurately extracted. This method is particularly suitable for identifying the degree of vegetation degradation due to construction, providing a quantitative basis for subsequent ecological restoration measures. For the assessment of soil erosion risk, LiDAR data can generate a three-dimensional terrain model, and key features such as slope and surface roughness can be automatically extracted using deep learning algorithms and combined with construction planning to identify high-risk areas prone to soil erosion.

In recent years, the development of object detection algorithms has significantly enhanced the accuracy and efficiency of image data extraction, particularly in the context of vegetation change monitoring and ecological restoration assessments [159,160,161,162]. Early approaches to extracting information from two-dimensional image data primarily relied on traditional image processing techniques, such as edge detection, histogram analysis, and template matching. While these methods were effective for basic image processing and object detection tasks, they faced limitations in robustness and efficiency when applied to complex scenes and large-scale data.

**Deep learning models:** Early object detection methods depended on handcrafted features and simple classifiers, such as Haar features and SVMs. These methods had limited detection efficiency and accuracy, especially in complex backgrounds. The advent of deep learning, particularly CNNs, marked a significant advancement in image processing. Deep learning models have excelled in object detection tasks. The YOLO (You Only Look Once) series of models is particularly notable for its real-time performance and efficiency, capable of detecting objects in a single forward pass. The evolution of YOLOv1 to YOLOv10 has continuously improved detection speed and accuracy [163,164,165,166,167,168,169,170]. The Single Shot MultiBox Detector (SSD) model employs multiple scales of prediction boxes to balance speed and accuracy, making it a crucial model for real-time object detection [171]. Faster R-CNN generates candidate boxes through a Region Proposal Network (RPN) and then uses CNNs for fine classification and regression [172,173]. Although it is slower than YOLO and SSD, Faster R-CNN has the advantage of higher detection accuracy. In response to the mechanization of transmission line engineering activities, improvements to the YOLOv5s model, such as the introduction of the GhostConv module in the backbone network to reduce the learning cost of non-essential features and the addition of a convolutional attention module in the Neck network to enhance the extraction of exposed soil features, are expected to provide critical support for early erosion hazard detection and help maintain the natural environment around transmission lines [174]. The introduction of ResNet (Residual Networks), which addresses the vanishing gradient problem in deep network training, has made it feasible to build very deep neural networks. ResNet has demonstrated exceptional performance in image classification tasks and is widely used in various environmental monitoring applications [175,176,177]. Furthermore, DenseNet (Densely Connected Convolutional Networks) establishes dense connections between each layer and all preceding layers, facilitating efficient feature reuse and significantly enhancing model efficiency and performance. DenseNet excels in processing high-resolution multispectral and hyperspectral images, making it suitable for fine classification of land cover features within construction zones [178,179]. Ref. [180] proposes a bird’s nest recognition method that combines visual saliency and deep learning, which not only has the advantage of rich feature information in visible images but also has the advantage of salient bird’s nest targets, which is instructive for the operation and maintenance of transmission lines. Ref. [41] used the Random Forest algorithm as a classifier model to analyze land use and land cover (LULC) changes in remotely sensed images of the construction site and its surroundings of the Akkuyu nuclear power plant project in Southern Turkey.

**Transformer models:** The introduction of Transformer models into image processing has demonstrated powerful feature learning and long-distance dependency modeling capabilities [181]. Detection Transformer (DETR) leverages the Transformer architecture to convert the object detection problem into a set prediction problem, effectively capturing target information in images through attention mechanisms without relying on region proposal networks [182,183,184]. The Vision Transformer (ViT) method segments images into fixed-size patches and inputs these patches as sequences into the Transformer, enabling efficient extraction of global features. ViT demonstrates significant advantages in processing large-scale remote sensing images and can be employed for precise classification and monitoring of extensive environmental changes, such as vegetation recovery and land cover classification [185,186].

**Multi-scale pyramid:** Feature Pyramid Networks (FPN) construct multi-scale feature pyramids by combining high- and low-resolution feature maps, enabling precise detection of objects at different scales [187,188]. FPNs are commonly used as foundational modules integrated into other object detection frameworks, such as Faster R-CNN and RetinaNet [189,190].

**Self-supervised models:** Self-supervised learning, which leverages unlabeled data for pre-training followed by fine-tuning on specific tasks, greatly improves model generalization. SimCLR (Simple Framework for Contrastive Learning of Visual Representations) uses contrastive learning to create positive and negative sample pairs through data augmentation, efficiently utilizing unlabeled data [191]; BYOL (Bootstrap Your Own Latent) further enhances feature learning efficiency by employing a self-supervised learning framework that does not require negative samples [192,193]. SimCLR can be used to pre-train models for identifying various environmental change patterns, such as vegetation degradation and water pollution [194].

However, these advanced network models require large amounts of high-quality annotated data. Moreover, existing deep learning models like YOLO, SSD, and Faster R-CNN need to be optimized for efficient processing of high-resolution, multispectral, and hyperspectral images. In complex environments, such as those with occlusions, lighting variations, and diverse surface features, the robustness of these models can decline, leading to higher rates of false positives and missed detections. The dynamic factors in complex environments, such as vegetation cover changes and construction equipment movement, may affect the robustness of these models under different conditions, leading to classification errors. Additionally, processing and classifying high-resolution, multispectral, and hyperspectral image data require substantial computational resources and storage space.

The primary types of three-dimensional data include point clouds and three-dimensional images. The rapid development of three-dimensional data image mining technologies has significantly enhanced the ability to monitor and assess complex terrains and structures.

For point cloud processing within three-dimensional image data, techniques such as the Iterative Closest Point (ICP) algorithm and the Normal Distributions Transform (NDT) algorithm (refs. [195,196]) have been employed to achieve precise alignment of multi-viewpoint cloud data, thereby enhancing the accuracy of monitoring data. For instance, the ICP algorithm is widely used in terrain surveys at construction sites due to its iterative optimization process that minimizes the distance error between two point clouds [197]. Feature extraction from point cloud data, using methods such as normal vector analysis, allows the identification and classification of different terrain features, such as planes, edges, and corners [198]. Point cloud segmentation into different regions or objects facilitates more detailed analysis and processing [199,200]. Segmentation techniques, including region growing and clustering-based methods, are particularly useful for identifying various structures and equipment during construction.

In processing three-dimensional images, voxelization—discretizing three-dimensional image data into voxels—enables volume calculation and spatial structure analysis [201]. Additionally, deep learning methods, such as PointNet and PointCNN, have been applied to point cloud data to automatically extract complex feature information, improving the precision and efficiency of point cloud data processing [202,203]. Representing three-dimensional data as graph structures and analyzing them using Graph Neural Networks (GNNs) allows for the capture of complex relationships and features within the data, aiding in a detailed assessment of the environmental impact of power transmission and distribution projects [204].

While three-dimensional data image processing technologies play a crucial role in environmental risk monitoring for power transmission and distribution projects, challenges remain in terms of data processing complexity and model training costs. Point cloud processing and three-dimensional image analysis require significant computational resources and complex algorithms, such as the ICP algorithm and multi-scale feature extraction techniques. The acquisition of three-dimensional data often involves technologies like LiDAR and multi-view imagery, which are characterized by high density and diversity.

### 5.3. Integration of Environmental Risk Data

The development of data fusion technologies has significantly enhanced the accuracy of monitoring results and the scientific rigor of decision-making in environmental risk monitoring during the construction phase of transmission and distribution projects. The impacts of environmental risks are usually determined by the overlap of multiple factors, and it is difficult to fully characterize them with a single data source. In the data fusion stage, we integrate data from multiple sources for comprehensive analysis. We reveal the dynamic change pattern of environmental risks during the construction cycle; we fuse image data and sensor data to improve the accuracy of risk identification. Data feature fusion supports the development of intelligent monitoring and early warning systems, providing an important technical guarantee for the accurate management of environmental risks during the construction period. Data sources for fusion include multispectral and panchromatic image integration, as well as the fusion of optical and radar images. The evolution of data fusion techniques has progressed from simple data overlay to more sophisticated multi-source data fusion methods. Below is a detailed explanation of the data fusion process and its technological development.

#### 5.3.1. Feature Dimensionality Reduction Method

In the context of environmental risk monitoring during the construction phase of power transmission and distribution projects, dimensionality reduction techniques play a crucial role in simplifying data structures, enhancing computational efficiency, and improving predictive accuracy in complex environmental scenarios. Principal Component Analysis (PCA) and Kernel Principal Component Analysis (KPCA) are widely applied in the environmental risk monitoring of construction projects.

PCA, a classical linear dimensionality reduction method, operates by projecting high-dimensional data onto a new lower-dimensional space through linear transformations, preserving the maximum variance of the data. This simplification of data structures removes redundant information, thereby speeding up model training and enhancing performance. PCA’s simplicity and high computational efficiency make it well-suited for linearly separable data [205,206].

On the other hand, KPCA extends PCA to non-linear contexts by employing a kernel function to map data into a higher-dimensional space, where PCA is then performed. KPCA captures the non-linear structures within the data, making it particularly effective for feature reduction in complex environments. For example, in the analysis of SAR imagery, KPCA has been utilized to reduce non-linear features, extracting key environmental change information to monitor vegetation changes and soil erosion [207].

In deep learning approaches, Autoencoders represent an unsupervised neural network architecture that encodes input data into a low-dimensional representation before decoding it to reconstruct the original data. Autoencoders can automatically learn the low-dimensional structures of data, making them suitable for large-scale data reduction. For instance, in vegetation restoration monitoring, Autoencoders have been employed to extract low-dimensional feature representations from high-resolution imagery data, effectively identifying and assessing ecological changes and environmental risks [208].

#### 5.3.2. Feature Selection Method

Feature Selection is the process of identifying the most relevant attributes while removing redundant and irrelevant ones, aiming to enhance the accuracy and efficiency of monitoring models. By selecting the most pertinent features, the performance of monitoring models can be significantly improved [209].

Filter-based methods and statistical approaches, such as analysis of variance (ANOVA) and Chi-square tests, evaluate the importance of each feature using statistical measures. For instance, in monitoring noise and dust data, ANOVA can be employed to filter out features with the greatest variance, thereby improving the model’s discriminatory power [210,211,212]. Embedded methods, lasso regression, by introducing the L1 norm as a regularization term, effectively address multicollinearity issues by shrinking the coefficients of less important features to zero, thus facilitating feature selection. In dust data analysis, for example, Lasso regression has been used to select the most relevant meteorological parameters, such as wind speed and humidity, enhancing the predictive accuracy of the model [213]. Decision trees and random forests evaluate feature importance based on the splitting nodes of the tree model [214,215]. In deep learning methods, CNNs automatically extract high-level features through convolutional layers and select the most important features during the training process. This approach is particularly suited for the analysis of image and video data. For instance, in vegetation cover change monitoring, a pre-trained CNN model has been employed to extract critical features from multispectral imagery, improving the accuracy and efficiency of monitoring [216]. The Recursive Feature Elimination (RFE) method recursively eliminates features that contribute least to the model’s performance, ultimately selecting the optimal subset of features and thereby enhancing the model’s predictive performance. In environmental monitoring during construction projects, RFE can be utilized to identify the environmental parameters that contribute most significantly to model prediction, such as soil moisture and vegetation cover [217].

#### 5.3.3. Feature Fusion Technology

Feature fusion techniques play a pivotal role in environmental risk monitoring during the construction phase of power transmission and distribution projects. By integrating features from various data sources—such as optical images, radar imagery, and multispectral data—these methods offer more comprehensive and precise environmental assessment tools.

Spatial Domain Methods, such as pixel-level fusion, directly combine raw information from source images. For instance, the fusion of low-resolution multispectral images with high-resolution panchromatic images can yield images that retain high-resolution detail while preserving multispectral information. This method is straightforward and effective for accurately monitoring ecological changes in construction areas [218]. Feature-level fusion involves extracting and integrating features such as edges, textures, and shapes from different data sources, effectively combining their strengths [219,220]. This technique enhances the ability to identify and classify environmental features like various vegetation types and land cover [221]. Additionally, decision-level fusion combines the processed features from different sources at a higher level, often using techniques like voting or weighted averaging to improve decision accuracy, providing a more comprehensive environmental risk assessment by synthesizing results from multiple data sources [222,223]. Frequency Domain Methods, techniques like Fourier transform, convert spatial data into the frequency domain for fusion, combining high-frequency and low-frequency information to enhance image detail and overall quality [224,225]. Wavelet Transform, through multi-scale decomposition and reconstruction, achieves data fusion that preserves both fine details and global information, making it well-suited for complex environments. This method is particularly effective in capturing terrain and vegetation changes in transmission and distribution project areas [226]. Deep Learning Methods, such as CNNs, utilize multiple layers of convolution and pooling operations to automatically extract and fuse features from data, such as the fusion of multispectral and panchromatic images. This approach significantly improves the accuracy of image classification and recognition [227]. GANs, through adversarial training between a generator and a discriminator, achieve data fusion that can provide high-quality image data in complex environments, offering strong support for environmental risk assessment in power transmission and distribution projects [228].

## 6. Construction of Risk Atlas

### 6.1. Technical Framework for Environmental Risks Atlas Construction

During the construction of transmission and distribution projects, the creation of environmental risks atlas primarily relies on the integration, processing, and analysis of multi-source data. The technical framework encompasses key stages such as data acquisition, preprocessing, feature extraction, data fusion, risk assessment, and map generation. Initially, various sensor devices, including ground sensors, drones, and satellites, are used to collect environmental factor data, such as tower base disturbances, vegetation changes, construction noise, and dust emissions. Subsequently, data preprocessing and cleaning are performed to ensure quality and consistency. Feature extraction techniques are then employed to identify and extract key features that characterize environmental risks. This is followed by the application of data fusion techniques, which integrate data from different sources to obtain comprehensive risk assessment information. Finally, based on the assessment results, environmental risk maps are generated, highlighting potential environmental risks and their spatial distribution during the construction of transmission and distribution projects.

A risk atlas is a knowledge organization based on a graph structure that transforms data into a computable knowledge network through nodes and edges. The environmental risk atlas and advance warning method based on the risk atlas achieves the modeling of implicit relationships and dynamic features in complex environmental systems by transforming heterogeneous data from multiple sources (such as monitoring equipment data, historical event data, geographic information, etc.), into a semantically linked knowledge network. It has the advantages of strong data integration capability, strong model interpretability, and cross-domain knowledge integration. In risk monitoring and advance warning, it can accurately identify environmental risks, dynamically monitor and trend analysis, and identify risk propagation paths. The risk atlas method shifts the environmental risk monitoring from traditional passive response to intelligent prediction, greatly improving the monitoring efficiency and accuracy of advance warning; it can be quickly adapted to different scenarios, such as water pollution monitoring, air quality advance warning, and environmental management of engineering construction, and promotes cross-domain application of environmental risk management.

### 6.2. Data Processing and Algorithms for Environmental Risk Factors

The processing of environmental risk factors is crucial during the atlas construction process. First, multi-source data undergo noise removal, radiometric correction, and geometric correction during preprocessing to ensure data consistency. Feature extraction algorithms, such as Convolutional Neural Networks (CNNs) and deep learning models, are utilized to extract key features from multispectral, radar, and LiDAR data, which reflect environmental risk factors like vegetation cover changes, soil erosion, noise, and dust distribution. In the data fusion phase, algorithms based on spatial domain, frequency domain, models, and machine learning methods are employed to integrate multi-source data into unified risk information. Finally, a comprehensive risk assessment is conducted, and the results are mapped to produce a comprehensive environmental risk monitoring map.

### 6.3. Visualization and Risk Presentation of the Atlas

The risk atlas, constructed based on environmental risk factors, is primarily used for environmental monitoring during the construction of transmission and distribution projects. The hierarchical structure of risk atlas is generally based on the classification of environmental risk levels, which can be defined according to the severity and impact range of the risks. Common risk levels include low, medium, and high risks, each corresponding to a specific risk level and displayed using different atlas layers. The choice of information display method directly affects the usability and readability of the risk atlas. From the classification, monitoring, and processing of initial environmental risk factors, the environmental risks monitoring atlas for transmission and distribution projects can be derived. As illustrated in Figure 5.

## 7. Risk Advance Warning

The environmental risks advance warning system plays a crucial role during the construction phase of transmission and distribution projects, ensuring the smooth progression of the project while minimizing negative impacts on the ecological environment. This system encompasses several key processes, including the integration and preprocessing of multi-source data, the development of intelligent risk identification and assessment models, and the design and implementation of a real-time advance warning system. By leveraging advanced data processing technologies and artificial intelligence algorithms, the system enables real-time monitoring, identification, and warning of potential risks, thereby enhancing response speed and accuracy. As illustrated in Figure 6. The environmental risk monitoring and advance warning method based on the risk atlas dynamically constructs a semantic knowledge network through the integration of multi-source data for the identification of risk sources, the inference of propagation paths, and the prediction of trends. In the process of implementation, data collection and cleaning, risk atlas construction and dynamic updating, inference model deployment, and visualization platform development need to be completed. In addition, through real-time data access and multi-level warning mechanisms, accurate monitoring and rapid response to environmental risks can be achieved. Combined with sensor networks and machine learning algorithms, this approach significantly improves the efficiency and intelligence of environmental risk monitoring, which can provide scientific support for decision-making and promote the deep integration of ecological protection and engineering construction.

During the construction phase of transmission and distribution projects, the environmental risks advance warning system must be capable of real-time monitoring, analysis, warning, and response to various environmental risk factors. These risk indicators include tower base disturbances, vegetation cover changes, soil erosion, residential home demolitions, construction noise and dust, waste management and resource utilization, impacts of engineering measures, temporary facility construction and management, ecological restoration efforts, etc. The selection of environmental warning indicators should be dynamically adjustable and optimized based on real-time monitoring feedback, ensuring they remain relevant to the current environmental conditions. For example, during the construction preparation stage, emphasis should be placed on residential demolitions, temporary facility construction, and vegetation cover changes. In contrast, during the construction phase, the focus should shift to construction noise and dust, tower base disturbances, and the impacts of engineering measures. In the ecological restoration phase, attention should be directed towards the application of measures such as vegetation cover restoration and soil conservation. By utilizing this system, advance warnings can be issued for potential environmental risks based on real-time monitoring data and historical data analysis results, allowing for the timely implementation of necessary countermeasures. Common approaches include statistical models, machine learning models, and deep learning models. For instance, using machine learning algorithms such as SVM and RF, noise and air pollutant concentration prediction models can be developed. When the advance warning system issues an alert, relevant departments and personnel must respond promptly and implement appropriate environmental protection measures. A well-established warning response mechanism is essential, with clear responsibilities and response procedures. For example, if noise monitoring values exceed the standard, the construction party should be immediately notified to implement noise reduction measures and simultaneously report to the relevant regulatory authorities. Detailed emergency plans should be formulated for different types and levels of environmental risks. For instance, to address dust pollution, measures such as water spraying for dust suppression and covering exposed surfaces can be employed [229,230] to prevent soil erosion and vegetation planting, and slope protection construction can be undertaken [231,232].

The risk atlas method can be deeply integrated with the e-Construction 2.0 platform of the State Grid Corporation of China, and data interaction can be achieved through standardized APIs (Application Programming Interface), and an environmental monitoring and advance warning module can be added to the platform, linking the existing GIS system and Internet of Things (IoT) functions, so as to enhance the platform’s intelligent and ecological management capabilities. This integrated approach can seamlessly combine dynamic risk analysis and construction management to achieve synergistic optimization of construction behavior and environmental protection, which not only improves the efficiency of risk management but also provides technical support for the construction of a green smart grid for the State Grid Corporation of China. At the same time, the method is expected to become an innovative technical standard, promoting its promotion and application in domestic and international power transmission and distribution projects and further consolidating its leading position in the industry.

The key to designing and implementing a real-time advance warning system lies in constructing an efficient and responsive system. By utilizing cloud platform architecture and Internet of Things technology, the system can collect, transmit, and process multi-source data in real-time, quickly identifying and assessing potential environmental risks. The warning information is conveyed to relevant departments through an intuitive interface and a rapid dissemination mechanism, ensuring timely response actions to effectively manage and mitigate environmental risks during the construction of transmission and distribution projects.

## 8. Conclusions

Based on the research findings presented earlier, we have identified key research opportunities in environmental monitoring and advance warning systems during the construction phase of transmission and distribution projects. Future studies should focus more intensively on real-time monitoring and advance warning of critical ecological risk factors during the construction process, such as tower base disturbances, changes in vegetation cover, and soil erosion. These factors have significant long-term impacts on local ecosystems, yet current research on them remains insufficient, particularly regarding their specific behaviors and potential impacts under varying environmental conditions. Therefore, future research needs to expand monitoring areas and extend the temporal scale of studies to fully capture these complex ecological processes.

Research should also increasingly include transmission and distribution projects in low- and middle-income regions, especially those located in ecologically sensitive areas. Such geographic diversity is crucial for developing broadly applicable risk assessment and advance warning models that can improve predictive accuracy under diverse environmental conditions. Future efforts should also focus on optimizing data fusion and processing technologies, particularly in the areas of multi-source data preprocessing, intelligent risk identification, and assessment model enhancement. This will significantly improve the real-time and precision capabilities of risk advance warning systems, better addressing complex environmental challenges. Through large-scale empirical studies across various environments and conditions, future research can establish standardized data processing procedures and protocols. These standards will help integrate advanced environmental monitoring and advance warning technologies into the comprehensive risk management framework of transmission and distribution projects, ensuring the ecological safety and sustainability of these projects.

## Figures and Tables

**Figure 1 sensors-24-07695-f001:**
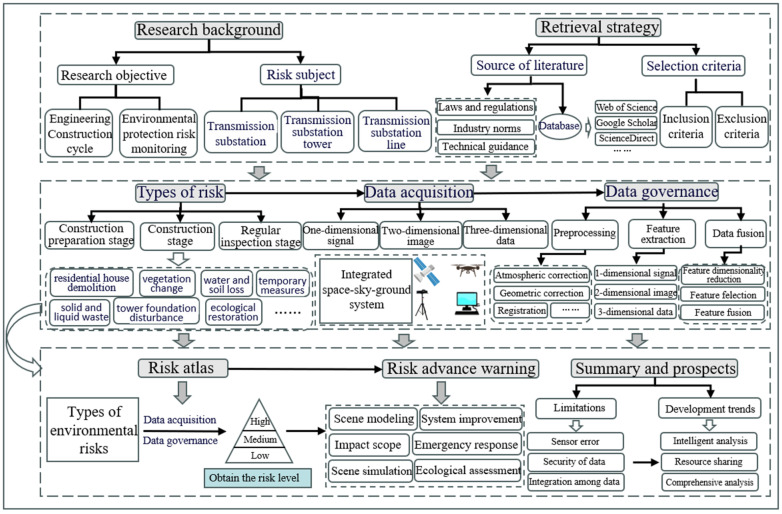
Article structure.

**Figure 2 sensors-24-07695-f002:**
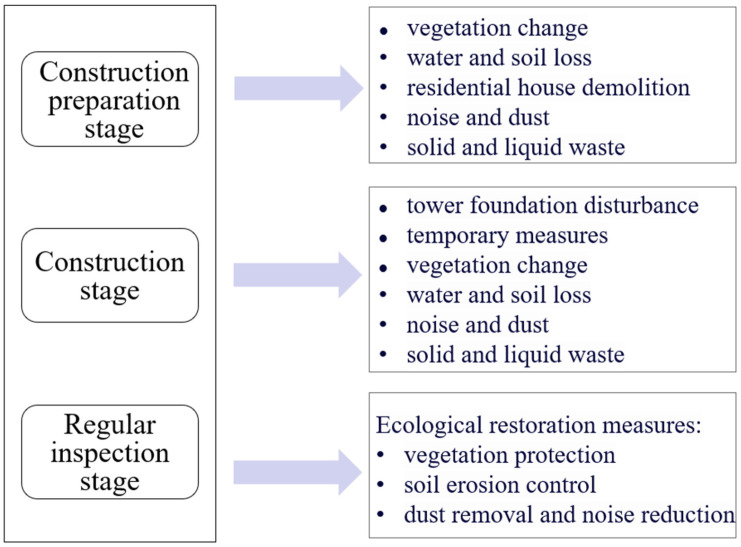
Types of environmental risks throughout the life cycle of transmission and distribution project construction.

**Figure 3 sensors-24-07695-f003:**
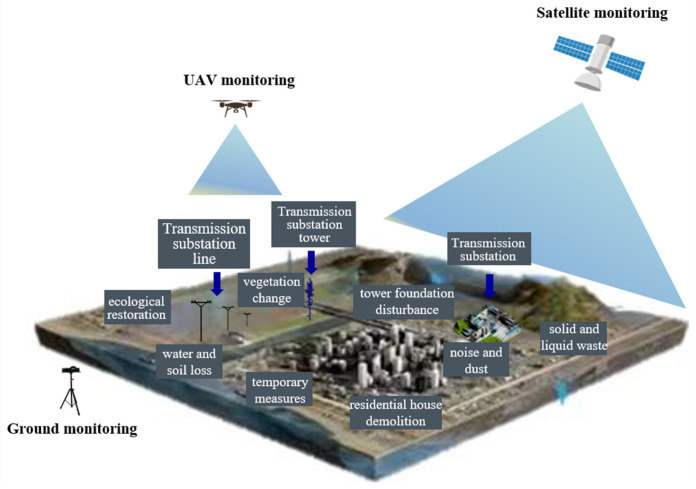
Integrated space–sky–ground system for environmental risks during the construction phase of transmission and distribution projects.

**Figure 4 sensors-24-07695-f004:**
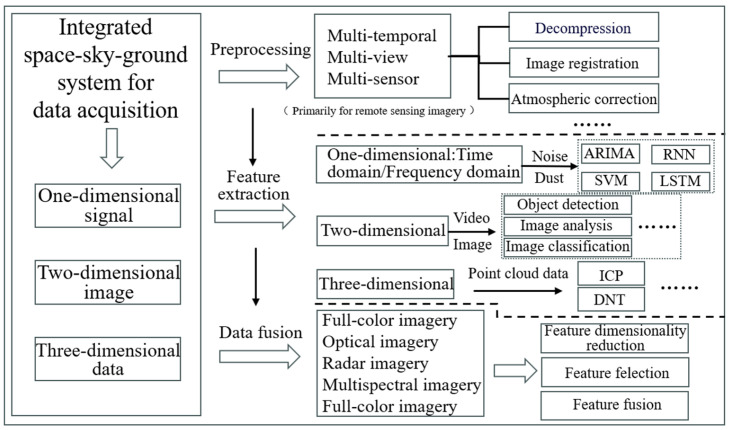
Integrated space–sky–ground system environmental risk factor data collection, processing, and analysis process.

**Figure 5 sensors-24-07695-f005:**
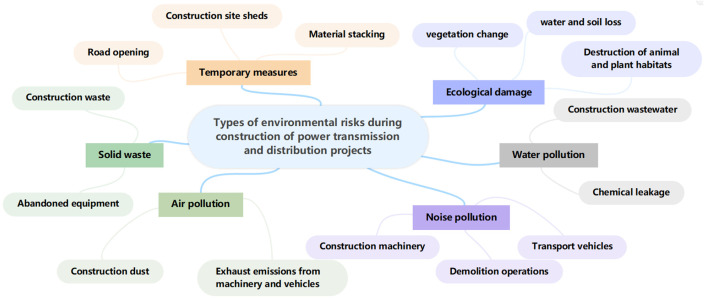
Environmental risk atlas during the construction of transmission and distribution projects.

**Figure 6 sensors-24-07695-f006:**
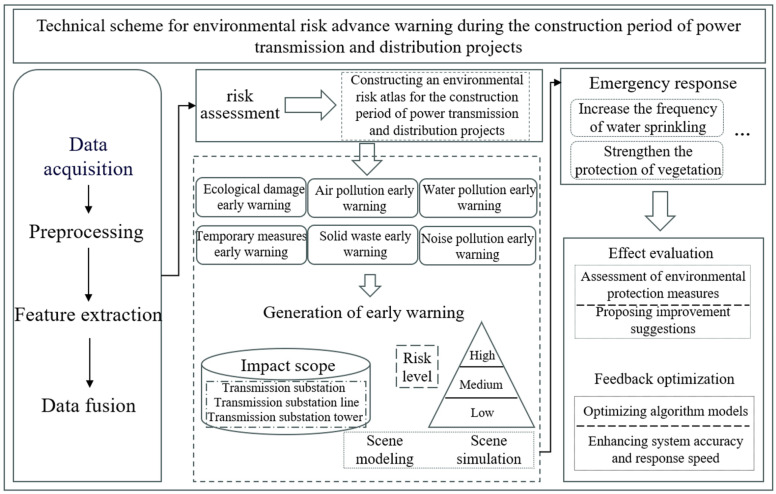
Technical scheme for environmental risks advance warning during the construction period of power transmission and distribution projects.

**Table 1 sensors-24-07695-t001:** Regulatory agencies and industry standards in the field of transmission and distribution projects (T&D Engineering) at the international level in the United States, the European Union, and China.

Country/Organization	Regulations/Guidelines
**International** International Finance Corporation (IFC) International Energy Agency (IEA) International Electrotechnical Commission (IEC)	Environmental, Health, and Safety Guidelines for Electric Power Transmission and Distribution (IFC, 2007) https://www.ifc.org/en/insights-reports/2000/ehs-guidelines-power (accessed on 17 July 2024) Transmission and distribution (T&D) (IEC, 2018) https://www.iec.ch/basecamp/transmission-and-distribution-td (accessed on 17 July 2024)
**The United States** American Electric Power (AEP) National Council on Electricity Policy (NCEP) U.S. Environmental Protection Agency (EPA) Electric Power Research Institute (EPRI)	Standards and Expectations for Siting, Real Estate, Right-of-Way, and Environmental Permitting for Transmission Interconnection Projects (AEP, 2022) https://www.pjm.com/planning/design-engineering/to-tech-standards/private-aep.aspx (accessed on 17 July 2024) Mini Guide on Transmission Siting: State Agency Decision Making (NCEP, 2021) https://pubs.naruc.org/pub/C1FA4F15-1866-DAAC-99FB-F832DD7ECFF0 (accessed on 17 July 2024) Guide for Successful Transmission Line Siting (EPRI, 2013) https://www.epri.com/research/products/3002001188 (accessed on 17 July 2024)
**The European Union** European Union (EU) European Commission (EC) European Environment Agency (EEA)	Transmission and distribution of electricity (EC) https://ec.europa.eu/sustainable-finance-taxonomy/activities/activity/295/view (accessed on 17 July 2024) Guidance on energy transmission infrastructure and EU nature legislation (EC, 2018) https://data.europa.eu/doi/10.2779/827210 (accessed on 17 July 2024)
**China** China Electricity Council (CEC) National Energy Administration (NEA) Ministry of Ecology and Environment (MEE)	Outline of Quality Supervision and Inspection of Power Transmission and Transformation Project (NEA, 2023) https://zfxxgk.nea.gov.cn/2023-05/08/c_1310731321.htm (accessed on 14 July 2024) Technical guidelines for environmental impact assessment of electric power transmission and distribution (MEE, 2020) https://www.mee.gov.cn/ywgz/fgbz/bz/bzwb/hxxhj/xgjcffbz/202012/t20201218_813935.shtm l(accessed on 14 July 2024) Technical requirements for environmental protection in electric power transmission and distribution construction project (MEE, 2020) https://www.mee.gov.cn/ywgz/fgbz/bz/bzwb/other/qt/202003/t20200305_767473.shtml (accessed on 14 July 2024) Technical specifications for environmental protection in electric power transmission and distribution for check and accept of completed project (MEE, 2014) https://www.mee.gov.cn/ywgz/fgbz/bz/bzwb/hxxhj/xgjcffbz/202012/t20201218_813936.shtml (accessed on 14 July 2024)

**Table 2 sensors-24-07695-t002:** Types of potential risks that engineering construction poses to the environment throughout the entire lifecycle of transmission and distribution project construction.

Stages	Activities	Ecological Influence
Construction preparation stage	Pre-construction preparations, including clearing the construction site, setting up temporary facilities, etc.	Vegetation destruction: The development of new roads in ecologically sensitive areas, such as mountainous regions and protected zones to facilitate the entry of equipment can lead to the destruction of surface vegetation. Trees may be cut down and vegetation may be damaged during the clearing of construction sites [1,2,9,10,11]. Soil erosion: Earthwork during the construction preparation phase may lead to soil erosion [3,4,5,6]. Residential demolition: The noise, dust, and waste generated from demolition activities may contribute to environmental pollution [7,12,13,14,15,16,17,18].
Construction stage	Infrastructure construction such as excavation and foundation work), structural installation (such as tower erection and cable laying), and substation equipment installation	Tower foundation disturbance: The excavation of foundation pits can lead to the destruction of vegetation and soil erosion, potentially triggering natural disasters such as landslides and debris flows [3,19,20,21]. Noise pollution: The noise generated by the operation of construction machinery and equipment may adversely affect neighboring residents and wildlife [7,22,23,24,25]. Air pollution: The dust and exhaust emissions produced during construction processes may have an impact on air quality [7,8,14,26,27]. Water pollution: Mud and construction wastewater generated during construction activities may contaminate local water bodies [3,28,29,30,31,32]. Waste disposal and resource utilization: The handling and recycling of construction waste generated, including concrete blocks, rubble, asphalt chunks, scrap metal, and other materials [22,32,33,34,35,36,37,38]. Soil and vegetation disruption: Temporary and engineering measures such as large-scale machinery operations and material stockpiling may further deteriorate soil structure and vegetation [28,39,40,41,42].
Regular inspection stage	Ecological restoration of disturbed sites	Ecological restoration of disturbed sites: This includes road rehabilitation, vegetation restoration measures, and soil erosion control efforts [20,42,43,44,45,46,47,48].

**Table 3 sensors-24-07695-t003:** The literature requirements for this study: databases, academic search tools, keywords, and number of documents, etc.

Databases/Academic Search Tools	Web of Science, IEEE Xplore, ScienceDirect, Ei Compendex, Google Scholar
Keywords	Power transmission and distribution projects, environmental protection risks, vegetation changes, soil erosion, UAV/satellite monitoring, risk atlas, feature extraction, data fusion, etc.
Number of Literature	An initial search yielded 1146 relevant English language publications. After applying inclusion and exclusion criteria, 232 English language articles were selected for further analysis.

**Table 4 sensors-24-07695-t004:** Total number of studies utilized in this review.

Databases	After Searching Keywords	Applying Selection Criteria
Web of Science	239	32
IEEE Xplore	145	28
ScienceDirect	226	41
Google Scholar	375	107
Ei Compendex	98	15
Others	63	9
Total	1146	232

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
