# Peer review of "Research on Environmental Risk Monitoring and Advance Warning Technologies of Power Transmission and Distribution Projects Construction Phase"

_sensors, 2024, doi:10.3390/s24237695_

Round 1

Reviewer 1 Report

Comments and Suggestions for Authors
  1. There is an extra comma before the word "Soil" in line 72, and an extra full stop after the word "health" in line 230. In the title of Table 1, "united states", "european union" and "china" should be corrected to "United States", "European Union" and "China" respectively.
  2. The content description from line 128 to line 150 lacks "Section 2".
  3. For "low-quality publications or venues" in line 167, it is advisable to provide quantitative criteria.
  4. The attributive description of "risk" in the title of line 181 is inconsistent with that in the article. It is recommended to be unified as "environmental risks".
  5. It is suggested that the title 3.3 be made consistent with the "regular inspection stage" in Figure 2 and Figure 1.
  6. In Section 4.3 regarding the three-dimensional signal, besides LiDAR, the Oblique photography method should also be mentioned. The former uses an active laser source while the latter utilizes natural light. Supplementary information is recommended.
  7. The description in Section 5.2 is recommended to be integrated. From the perspective of computer data processing, essentially both three-dimensional and two-dimensional data are processed by reducing them to one-dimensional data. It is proposed to summarize and describe it.

Author Response

Response to the 1st Reviewer

Comments 1: There is an extra comma before the word "Soil" in line 72, and an extra full stop after the word "health" in line 230. In the title of Table 1, "united states", "european union" and "china" should be corrected to "United States", "European Union" and "China" respectively;

Response 1: Thank you very much for your great support of our work, you have provided us with very valuable suggestions to improve the quality of this article! We acknowledge that we have made errors in the use of some words and punctuation in the article, and therefore we have made the following changes to the suggestions made by the reviewers:

  • We have amended “There is an extra comma before the word "Soil" in line 72”, which appears in line 79 of the manuscript.
  • We have amended “There is an extra full stop after the word "health" in line 230”, which appears in line 253 of the manuscript.
  • We have amended “In the title of Table 1, "united states", "european union" and "china" should be corrected to "United States", "European Union" and "China" respectively”, which appears in lines 95 to 96 of the manuscript.

Comments 2: The content description from line 128 to line 150 lacks "Section 2";

Response 2:Thank you for pointing this out. We agree with this comment. We have therefore amended this comment, which appears in line 150 of the manuscript.

Comments 3: For "low-quality publications or venues" in line 167, it is advisable to provide quantitative criteria;

Response 3: Thank you for your feedback! We have carefully read the content of line 167, and in conjunction with what we have given in Section 2, we have provided quantitative criteria for low-quality publications or venues, which appear in lines 188 to 189 of the manuscript, and in Table 3.

Comments 4: The attributive description of "risk" in the title of line 181 is inconsistent with that in the article. It is recommended to be unified as "environmental risks";

Response 4: Thank you again for your feedback! We have proofread and revised the attributional description of “risk”, which appears in line 202 of the manuscript (section 3 heading), and we have revised it in its entirety based on your suggestions. For the exact meaning of “environmental risks”, refer to the published literature (Fan et al., 2024).

[1] Fan X, Liu M, Zhang B, et al. The Green Premium: Environmental Regulation, Environmental Risk and Property Value[J]. Environmental and Resource Economics, 2024, 87(5): 1061-1096.

Comments 5: It is suggested that the title 3.3 be made consistent with the "regular inspection stage" in Figure 2 and Figure 1;

Response 5: Thank you very much for pointing this out! We acknowledge that this issue was overlooked in the submitted manuscript. As a result, we have revised the content of the relevant heading so that it appears consistently before and after, and it appears on line 295 of the manuscript.

Comments 6: In Section 4.3 regarding the three-dimensional signal, besides LiDAR, the Oblique photography method should also be mentioned. The former uses an active laser source while the latter utilizes natural light. Supplementary information is recommended;

Response 6: Thank you very much for your suggestion! We have consulted a number of sources to add to the article. We have therefore added the Oblique Photography method to the paper, which appears in lines 483 to 489 of the manuscript.

Comments 7: The description in Section 5.2 is recommended to be integrated. From the perspective of computer data processing, essentially both three-dimensional and two-dimensional data are processed by reducing them to one-dimensional data. It is proposed to summarize and describe it;

Response 7: Thank you for providing valuable suggestions! This section has merged the three data types, highlighting technical advances in data governance. See lines 630 to 777 in the text for more details, with advances in two-dimensional data governance methods in lines 616-738 and three-dimensional data governance methods in lines 739-777.

Reviewer 2 Report

Comments and Suggestions for Authors

This paper systematically reviews environmental risk monitoring and early warning technologies during the construction phase of power transmission and distribution projects. It summarizes current technical methods and proposes a knowledge-atlas-based approach to environmental risk early warning, offering significant guidance for monitoring and mitigating the environmental risks associated with transmission and distribution projects. Additionally, while the manuscript covers a broad range of research content comprehensively, certain sections lack complete clarity in expression. Specific details are as follows:

Comment1: Environmental risks arise during the preparation, implementation, and operation and maintenance (O&M) phases of construction. In the O&M phase, the aspects overlapping with the focus of this study are ecological restoration and regular inspections. The current manuscript lacks a comprehensive summary of the technical methods, status, and issues related to regular inspections, which requires further supplementation and improvement;

Comment2: The knowledge-atlas-based approach to environmental risk monitoring and early warning is one of the central perspectives of this paper. Compared to other methods, what are the core principles and advantages of this approach, what role can it play in risk monitoring and early warning, and what kinds of breakthroughs might it lead to? The author is encouraged to elaborate on these points. Additionally, the current explanation and description of the knowledge atlas concept presented in the paper are insufficient and require further elaboration;

Comment3: The ultimate goal of environmental risk monitoring is early warning. This paper proposes a technical method, but guidance on how to implement this method in practice is needed. Additionally, it is necessary to explore whether this method can be integrated with relevant platforms of the State Grid Corporation of China(e.g., e-Construction 2.0) to enable intelligent monitoring and early warning;

Comment4: Please provide a supplementary definition of one-dimensional signals, as many one-dimensional signals also contain two-dimensional information when considering time series. A brief clarification is needed to prevent potential misunderstandings;

Comment5: The content on data governance for environmental risk factors has weak relevance to the power grid. Additional information is needed to strengthen this connection.

This paper has a certain guiding role in the development of environmental risk monitoring and early warning technology for power transmission and distribution projects, especially the cross-fertilisation of remote sensing and grid technology, which can improve the existing level of monitoring and early warning to a certain extent, and the content is more in line with the scope of this paper's research, and it is agreed to be published after in-depth modification of this paper.

Comments on the Quality of English Language

The overall quality of English is good.

Author Response

Response to the 2nd Reviewer

Comments 1: Environmental risks arise during the preparation, implementation, and operation and maintenance (O&M) phases of construction. In the O&M phase, the aspects overlapping with the focus of this study are ecological restoration and regular inspections. The current manuscript lacks a comprehensive summary of the technical methods, status, and issues related to regular inspections, which requires further supplementation and improvement;

Response 1: We are very grateful for your professional review work on our article! We neglected to include in the text a study of the relevant technical approach and current status of the regular inspection phase, so we have added this, which appears in lines 318 to 328 in the manuscript.

Comments 2: The knowledge-atlas-based approach to environmental risk monitoring and early warning is one of the central perspectives of this paper. Compared to other methods, what are the core principles and advantages of this approach, what role can it play in risk monitoring and early warning, and what kinds of breakthroughs might it lead to? The author is encouraged to elaborate on these points. Additionally, the current explanation and description of the knowledge atlas concept presented in the paper are insufficient and require further elaboration;

Response 2: Thank you for your expert advice on how to put the finishing touches on the article. We have reorganised the article according to your comments and revised it as follows. See lines 895 to 909 of the manuscript for details.

  • Definition: Risk atlas is a knowledge organisation based on a graph structure that transforms data into a computable knowledge network through nodes and edges.
  • Core principles and advantages: The environmental risk atlas and advance warning method based on risk atlas achieves the modelling of implicit relationships and dynamic features in complex environmental systems by transforming heterogeneous data from multiple sources (such as monitoring equipment data, historical event data, geographic information, etc.) into a semantically linked knowledge network. It has the advantages of strong data integration capability, strong model interpretability and cross-domain knowledge integration.
  • Role and Contributions: In risk monitoring and advance warning, it can accurately identify environmental risks, dynamically monitor and trend analysis and identify risk propagation paths. The risk atlas method shifts the environmental risk monitoring from traditional passive response to intelligent prediction, greatly improving the monitoring efficiency and accuracy of advance warning; it can be quickly adapted to different scenarios, such as water pollution monitoring, air quality advance warning and environmental management of engineering construction, and promotes cross-domain application of environmental risk management.

Comments 3: The ultimate goal of environmental risk monitoring is early warning. This paper proposes a technical method, but guidance on how to implement this method in practice is needed. Additionally, it is necessary to explore whether this method can be integrated with relevant platforms of the State Grid Corporation of China(e.g., e-Construction 2.0) to enable intelligent monitoring and early warning;

Response 3: We thank you for your professional review work and valuable advice on our manuscript! We accept this suggestion. There is an omission in the elaboration of how to guide the implementation of advance warning of environmental risks in our manuscript, and based on your comments, we make the following changes, as detailed in lines 945 to 956 of the manuscript and lines 992 to 1005 of the manuscript.

  • Practical implementation: The environmental risk monitoring and advance warning method based on risk atlas dynamically constructs a semantic knowledge network through the integration of multi-source data for the identification of risk sources, the inference of propagation paths and the prediction of trends. In the process of implementation, data collection and cleaning, risk atlas construction and dynamic updating, inference model deployment and visualisation platform development need to be completed. In addition, through real-time data access and multi-level warning mechanisms, accurate monitoring and rapid response to environmental risks can be achieved. Combined with sensor networks and machine learning algorithms, this approach significantly improves the efficiency and intelligence of environmental risk monitoring, which can provide scientific support for decision-making and promote the deep integration of ecological protection and engineering construction.
  • Relation with e-Construction 2.0: The risk atlas method can be deeply integrated with the e-Construction 2.0 platform of the State Grid Corporation of China, and data interaction can be achieved through standardised APIs (Application Programming Interface), and an environmental monitoring and advance warning module can be added to the platform, linking the existing GIS system and Internet of Things (IoT) functions, so as to enhance the platform's intelligent and ecological management capabilities. management capability. This integrated approach can seamlessly combine dynamic risk analysis and construction management to achieve synergistic optimisation of construction behaviour and environmental protection, which not only improves the efficiency of risk management, but also provides technical support for the construction of green smart grid for the State Grid Corporation of China. At the same time, the method is expected to become an innovative technical standard, promoting its promotion and application in domestic and international power transmission and distribution projects, and further consolidating its leading position in the industry.

Comments 4: Please provide a supplementary definition of one-dimensional signals, as many one-dimensional signals also contain two-dimensional information when considering time series. A brief clarification is needed to prevent potential misunderstandings;

Response 4: Thank you very much for your great support of our work, you have provided us with very valuable suggestions to improve the quality of this paper! We fully understand your concern about the definition of one-dimensional signals in the manuscript. We have provided additional definitions and explanatory notes on one-dimensional signals in the manuscript, as detailed in lines 350 to 360 of the manuscript.

  • Definition: The one-dimensional signal is a scalar function that varies with an independent variable (such as time or space), and its core property is to record the dynamics of a single variable in a certain dimension.
  • Declaration in our study: In our study, the one-dimensional data involved are mainly time-series data of noise and dust, which are mainly concerned with the intensity or value that changes over time in the time dimension, such as the decibel value of noise and the concentration of dust. However when the time series is analysed in conjunction with additional characteristic variables (such as spatial location, wind speed, ambient temperature, etc.), the signal can essentially be mapped to a two or even higher dimensional characteristic space, such as when we analyse the correlation between the dust concentration with the wind speed or temperature and humidity in a time series, the data can be represented as a two or even multi-dimensional signal.

Comments 5:The content on data governance for environmental risk factors has weak relevance to the power grid. Additional information is needed to strengthen this connection;

Response 5:Thank you for your suggestions. We tried our best to improve the manuscript and made some changes to it. In our previous manuscript, we only described the state of the art and progress of data governance methods, which led to the weakening of the relevance of this part to the power grid, based on this, we have added a supplementary description of environmental factor data governance during the construction period of transmission and distribution projects and related examples in the manuscript, and we hope that this change can better show the role of environmental factor data governance in the construction of a green power grid, the details of the supplementary description can be found at For details, see lines 635 to 644 and lines 781 to 789 of the manuscript. For examples, see lines 606 to 610 and lines 684 to 690 of the manuscript.